# Precarious Employment and Chronic Stress: Do Social Support Networks Matter?

**DOI:** 10.3390/ijerph19031909

**Published:** 2022-02-08

**Authors:** Francesc Belvis, Mireia Bolíbar, Joan Benach, Mireia Julià

**Affiliations:** 1Research Group on Health Inequalities, Environment, and Employment Conditions (GREDS-EMCONET), Department of Political and Social Sciences, Universitat Pompeu Fabra, 08005 Barcelona, Spain; mireia.bolibar@ub.edu (M.B.); joan.benach@upf.edu (J.B.); mireia.julia.perez@psmar.cat (M.J.); 2Johns Hopkins University—Universitat Pompeu Fabra Public Policy Center (UPF-PPC), 08005 Barcelona, Spain; 3Department of Sociology, Universitat de Barcelona, 08034 Barcelona, Spain; 4Ecological Humanities Research Group (GHECO), Universidad Autónoma de Madrid, 28049 Madrid, Spain; 5ESIMar (Mar Nursing School), Parc de Salut Mar, Universitat Pompeu Fabra-Affiliated, 08005 Barcelona, Spain; 6SDHEd (Social Determinants and Health Education Research Group), IMIM (Hospital del Mar Medical Research Institute), 08005 Barcelona, Spain

**Keywords:** chronic stress, precarious employment, social support networks, buffering hypothesis, cortisol, stress biomarkers, health inequalities, social determinants of health

## Abstract

Precarious employment has been identified as a potentially damaging stressor. Conversely, social support networks have a well-known protective effect on health and well-being. The ways in which precariousness and social support may interact have scarcely been studied with respect to either perceived stress or objective stress biomarkers. This research aims to fill this gap by means of a cross-sectional study based on a non-probability quota sample of 250 workers aged 25–60 in Barcelona, Spain. Fieldwork was carried out between May 2019 and January 2020. Employment precariousness, perceived social support and stress levels were measured by means of scales, while individual steroid profiles capturing the chronic stress suffered over a period of a month were obtained from hair samples using a liquid chromatography-tandem mass spectrometry methodology. As for perceived stress, analysis indicates that a reverse buffering effect exists (interaction B = 0.22, *p* = 0.014). Steroid biomarkers are unrelated to social support, while association with precariousness is weak and only reaches significance at *p* < 0.05 in the case of women and 20ß dihydrocortisone metabolites. These results suggest that social support can have negative effects on the relationship between perceived health and an emerging stressful condition like precariousness, while its association with physiological measures of stress remains uncertain.

## 1. Introduction

Social support can be defined as the supportive relationships that arise with friends, family members, and others [1]. The relationship between social support levels, stressor events (or a stressful environment), and health or wellbeing outcomes has been studied largely by behavioural and medical scientists, under the assumption that social support provides some kind of protection against the psychological or physiological disorders produced by chronic stress. One of the remaining research questions on this issue is the specific nature of this relationship, where the main effects model (that is to say, stressor and social support acting independently on health status, negatively in the first case and positively in the second) is often opposed to the “buffering hypothesis” [2]. While the buffering hypothesis posits that the positive relationship between social support and health is due, only or primarily, to the protection that social support offers persons from the potentially pathogenic influence of stressful events, the main effects model proposes that social resources have a beneficial effect irrespective of whether persons are under stress. Both theories also differ in respect of the pathways through which participation in social networks can affect psychological well-being [3].

The relationship between stressful events and social support has also been extensively tested for stressful work-related situations (in relation to both the main effects model and the buffering hypothesis paradigm) [4], such as unemployment [5,6,7,8], job searches [9], or intense work [10], but it has only very recently been tested for employment precariousness [11]. Although there is still no full consensus on its definition, precarious employment might be considered a multidimensional construct encompassing dimensions such as employment insecurity, individualized bargaining relations between workers and employers, low wages and economic deprivation, limited workplace rights and social protection, and powerlessness to exercise workplace rights [12]. In recent decades, and also due to different economic crises, technological changes, and policies that have weakened unions’ collective bargaining power, there has been an increase in the flexibility of employment conditions with profound changes that have led to the precarization of employment conditions [13,14]. Employment precariousness is particularly prevalent among the most vulnerable social groups in the labor market, like women, immigrants, the working class, and young workers. Moreover, early evidence suggests that the recent COVID-19 pandemic has increased precarious employment and that precarious workers will we exposed to serious stressors [15].

Certainly, precarious employment has been identified as a relevant social determinant of health and health inequalities that has a strong impact on self-perceived health, including mental health [12]. Among the pathways and mechanisms linking these two outcomes, several are related to increased stress. Workers under situations of precarious employment often face greater demands or have less control over the work process and experience social isolation [16] and a lack of support [17]. These experiences have been identified as powerful social stressors [16], which in turn can be linked to adverse health and well-being outcomes [18]. More generally, precarious jobs may limit workers’ control over their professional and personal lives, leading to experiences of job insecurity, feelings of betrayal and injustice, feelings of powerlessness and being out of control, a lack of future opportunities, or a lack of professional identity [19,20,21,22]. As for precarious employment, there is a suspicion that its relation to health outcomes may be affected by a deleterious impact on the social support networks [23]. Individuals in a more precarious situation tend to have smaller and less diverse networks [24] and, given the tendency to socialize with people from similar socio-economic situations [25], the networks of those in a precarious situation also tend to be less capable of providing resources. This will not only result in lower levels of social support for precarious workers, but also the buffering capacity of the support received will be eroded.

A second gap in social epidemiological research on work stress events and social support networks is that it is mostly focused on perceived measures of health. This may be a consequence of the narrow division among disciplines that has led researchers to consider social causes aside from the biological ones together with the difficulties in collecting large-scale reliable biological data [26]. This has begun to change and in recent years, much attention has been given to the study of hair cortisol as a biomarker of chronic stress. Cortisol is a glucocorticoid hormone released after stimulation of the hypothalamic-pituitary-adrenal axis (HPA), which is activated by stress and is involved in the body’s response to stress [27]. In contrast to other sources such as urine, saliva, and plasma, which provide one-off momentary information about the individual’s hormonal status and are highly affected by different factors such as the circadian rhythm or fear of needles, determining the amount of cortisol in hair would provide information about the chronic status of the steroid hormones [28].

Literature specifically regarding the relationship between social support and hair cortisol is extremely scarce, and the results are also contradictory. There is a doctoral thesis [29] that is dedicated to studying the relationship of psychosocial factors and high cortisol concentrations (HCC), including social support measured in three dimensions: its assessment, feelings of belonging, and tangible support. Using a sample of 165 workers, the relationships were found to be inverse to what was hypothesized by the author herself, given that greater social support has been directly related to higher levels of HCC. The study by [30] also considers the role of social support as a buffer against stressful situations. Using a sample of 71 patients with bipolar disorder, the authors first positively correlated the number of negative events that occurred in the subjects’ life with cortisol levels in the latter’s hair, showing how this relationship is totally minimized by self-perceived instrumental and emotional social support.

Considering the scarcity of studies and uncertainties highlighted in this section, the objective of this paper is twofold: (a) to test the main effect and possible moderator role of social support on the relationship between precarious employment, understood as a stressful factor, and perceived stress levels, and (b) to apply the same analytical framework to biological markers of chronic stress.

## 2. Materials and Methods

### 2.1. Study Design, Sample and Variables

This is a cross-sectional study corresponding to the second phase of a three-phase sequential mixed research design oriented to identify the pathways and mechanisms explaining the well-established relationship between precarious employment and health [23]. In the previous phase, secondary analysis of the Survey on Workers and the Unemployed of Barcelona (EPYPB, 2017–2018) was used to map the sociogeographical distribution of precarious employment in the city of Barcelona. This information was used to implement a non-probabilistic sampling strategy of *n* = 255 cases based on proportional quotas based on sex, age group, place of origin (born in Spain vs. born abroad), and the socioeconomic level of the district of residence (medium, medium-high, or high vs. medium-low and low-income districts). Due to practical and financial constraints, studies involving hair cortisol are unlikely to consist of large samples. The overall sample sizes in a systematic review of studies addressing hair cortisol and chronic stress or mental health relationship range from under *n* = 100 to *n* = 395 at a maximum [28], while the sample in [31] is unusually large (*n* = 760). Therefore, our sample was designed to show how cortisol in hair (and other related biomarkers of the HPA axis) are distributed throughout the whole range of experiences of precarious employment previously identified in Barcelona and to unravel the explanatory power of the pathways and mechanisms linking precariousness to health, in particular, social support. Nonetheless, under the assumptions of an alpha risk of 0.05 and a beta risk below 0.2 (80% power) in a two-sided test and with a sample loss rate of 0%, our sample size will allow us to estimate correlation coefficients of 0.175. Participants were recruited from the pool of participants in the EPYPB within the selected age range of people who had agreed to being contacted again for future studies and specifically for this project (*n* = 1210). Also, in order to offset the bias of this subsample toward profiles with higher levels of education and income, the abovementioned recruitment strategy was complemented with 40 individuals contacted through social and labor organizations that work with groups of precarious workers, with a slight overrepresentation of immigrant women. Inclusion criteria were: (i) being a salaried worker or bogus self-employed worker serving a single employer, (ii) being between 24 and 60 years old, (iii) living independently in Barcelona (i.e., young people living with their parents were excluded), (iv) the length of hair at the back of the head being at least one centimeter, and (v) not having taken holidays within the month prior to the interview. Exclusion criteria were: (i) having taken corticosteroids within the month prior to the interview, (ii) being diagnosed with an adrenal disease, and (iii) being pregnant, due to possible alterations in cortisol levels.

The questionnaire collected information about the main concepts involved in the present analysis, including the sociodemographics used as control variables. Employment precariousness was measured by means of the Precarious Employment Scale (EPRES), a validated instrument [32]. As for social support, the Duke-UNK-11 Functional Social Support scale [33] was used. This is a perceived social support scale that has the advantages of being multidimensional, standing out for its simplicity and brevity, and having also been validated in Spain [34]. Perceived stress levels were determined by means of the Perceived Stress Scale (PSS), a 14-item measure of experienced levels of stress [35]. Upon completing it, a sample of participants’ hair equivalent to a lock of hair of the thickness of a pen (between ~30 and 50 mg) was taken by the previously trained interviewers from the back of the head using scissors cutting as close to the skin as possible. The first centimeter of the lock of hair that is in contact with the scalp is the biological material subjected to laboratory analysis. Since hair grows about one centimeter per month, the selection of this segment implies that the level of chronic stress accumulated over the month prior to sampling can be identified. The steroids profile (simultaneous levels of cortisol and other steroid metabolites) was measured in hair samples through a validated protocol based on liquid chromatography-tandem mass spectrometry (LC-MS/MS) [36]. Briefly, samples were washed with dichloromethane and shredded with a ball mill. After weighting (c.a. 50 mg) and adding the internal standard, steroids were extracted with methanol. Analytes were preconcentrated using a liquid–liquid extraction with ethyl acetate and determined by LC-MS/MS using an Acquity UPLC system (Waters Associates, Milford, MA, USA) coupled with a triple quadrupole (TQS Micro) mass spectrometer provided with an orthogonal Z-spray-electrospray interface (ESI) (Waters Associates). Steroids were quantified by external calibration using labelled steroids as internal standards. Concentrations, normalized by the weight for each sample, were expressed as nanogram steroid/milligram hair (ng/mg). Up to seven biomarkers of the HPA axis, which plays a central role in the physiological response to stress, were considered in this research: Cortisol and their metabolites 20α-dihydrocortisol (20αDHF) and 20ß-dihydrocortisol (20βDHF); Cortisone and their metabolites 20α-dihydrocortisone (20αDHE) and 20ß-dihydrocortisone (20βDHE); and A_11dehydrocorticosterona (11-DHC). Many of these substances are closely related isomers. Cortisone is a metabolite from cortisol with a similar function and also correlated with stress levels in the subject, and A_11dehydrocorticosterona is an endogenous corticosteroid related to cortisone and corticosterone.

### 2.2. Statistical Analysis

Variables in the research and their relationships were described by means of univariate and bivariate statistics, and their statistical properties were investigated. Biomarker variables were log-transformed in order to normalize their highly skewed distributions.

A linear model predicting perceived stress from precariousness and social support levels was adjusted to the data, with sex and age group as control variables. The former main effect model was compared to a second one allowing for an interaction between precariousness and social support to test the existence of a buffering effect. Interaction effects were studied by graphical means, and the Johnson-Neyman method [37,38] was used to establish the range of values of social support within which precariousness significantly impacts stress levels.

Similar models were adjusted for each of the available biomarkers. In this case, however, Body Mass Index (BMI) was added as a control variable, as evidence exists that body weight partly determines these biological outcomes [39]. Moreover, an additional interaction between sex and precariousness levels was allowed, as bivariate analysis stratified by sex (Figure A1) suggested that the relationship between precariousness and some biomarkers was stronger in the case of women.

The main analyses were performed with the R environment and the R version of the PROCESS macro v.4.0 [40]. Interactions were further investigated with the homonymous R package [41], and the linear model regression assumptions were tested by means of the gvlma package based on [42].

## 3. Results

### 3.1. Descriptives

Information for the *n* = 255 individuals in the sample is almost complete in the variables considered except for a few cases in some biomarkers (Table 1). Observation of ranges and quartiles reveals that the distribution of biomarkers is highly positively skewed.

The analysis of bivariate correlations between the variables in this study shows very clear patterns: (a) precariousness, social support, and perceived stress are moderately correlated with one another; (b) biomarkers are strongly positively correlated with one another; (c) the former groups are virtually unrelated, with the exception of precariousness, which is weakly positively correlated with some biomarkers; and (d) control variables show scattered weak positive and negative correlations with groups of variables (a) and (b) (Figure 1).

### 3.2. Perceived Stress

Table 2 shows the main effects (1) and interaction (2) regression results of social support and employment precariousness predicting perceived stress levels. Estimated coefficients are very similar, and for this reason we will only comment on model (2) outcomes. The main effect of EPRES is positive and significant (B = 4.32, *p* < 0.001), indicating that higher levels of precariousness are associated with higher stress, while the main effect of social support is negative and significant (B = −0.27, *p* < 0.001), meaning that higher social support is related to lower levels of stress. The interaction coefficient of the former factors is also significant and positive (B = 0.22, *p* < 0.014) and represents an increase of 0.0151 in the determination coefficient (R^2^) of model (2) respect model (1). Additionally, being a woman significantly increases stress levels (B = 3.45, *p* < 0.001), while being above the age of 34 years old significantly decreases them (B = −2.27, *p* < 0.019). Overall the predicting capacity of the model (R^2^ = 0.304) is acceptable, and no significant departures from linear regression assumptions were found (Table 2).

Further analyses have been carried out on the interaction between employment precariousness and social support. Figure 2a shows that the impact of the precariousness level on the stress level decreases as social support increases for all levels of precariousness, but the moderator effect of social support is greater at low precariousness levels (−1 SD, dotted blue line) than at the mean (0 SD, dashed violet line) or high levels of precariousness (1 SD, solid red line). Specifically, the slope of DUFSS is B = −0.391, *p* < 0.001 in the case of low precariousness, B = −0.270, *p* < 0.001 in the case of medium precariousness, and B = −0.149, *p* < 0.05 in the case of high precariousness.

According to the Johnson–Neyman method (Figure 2b), the effect of precariousness on stress is significantly positive (*p* < 0.05) outside the region determined by the (centered) values of social support [−96.55, −9.53]. As can be seen, most of the observed cases in our sample (87.5%) are above this −9.53 value of social support (vertical blue dashed line).

### 3.3. Biomarkers of Chronic Stresss

The interaction between precariousness and social support unveiled in the perceived stress analysis is completely absent when biomarkers are used as an outcome in the regressions (Table 3 and Table A1 for 20αDHE biomarker).

In fact, there is not a single indication that social support influences biomarkers at all. As for Cortisol, neither the main effect nor the interactions are significant for the predictors considered, and the whole model is useless (R^2^ = 0.017, *p* = 0.7984). The BMI is the only significant predictor in the case of 20αDHF (B = 0.05, *p* = 0.001), while being a woman predicts 20βDHF and cortisone levels (B = −0.17, *p* = 0.015 and B = −0.23, *p* = 0.001, respectively). Higher BMI, being a woman, and being older are also negatively associated with 11-DHC levels (B = −0.17, *p* = 0.009; B = −0.19, *p* = 0.018 and B = −0.02, *p* = 0.014, respectively). Finally, there is little evidence that precariousness influences 20αDHE and 20βDHE biomarkers levels, as the interaction between EPRES and sex appears to be significant (B = 0.24, *p* = 0.071 Table A1, and B = 0.24, *p* = 0.042, respectively).

Further analyses have been carried out concerning the interaction between sex and precariousness on the 20βDHE biomarker, in which this interaction was significant at *p* < 0.05. Figure 3 shows that 20βDHE levels are practically unaffected by precariousness in men (blue dashed line), whereas a positive association does exist for women (red solid line). Specifically, the slope of EPRES in the case of men is B = 0.017, *p* = 0.850, whereas in the case of women it is B = 0.257, *p* < 0.01.

## 4. Discussion

In this paper, we have studied the effect of employment precariousness and social support on both perceived stress and steroid profiles. As for the first outcome, expectations that perceived stress will be associated positively with precariousness and negatively with social support have been confirmed. This result is in line with most cross-sectional research that has tested the main effects of stressors and social support on health outcomes [43] and work-related stress events and social support specifically [5,11]. We have also found that workers who are more precarious have lower levels of social support than less precarious ones (i.e., both factors are negatively correlated). Moreover, the overall effect of precariousness and social support on perceived stress is not simply summative, given that a significant interaction effect between the two factors was found. However, the interaction has little explanatory power and, more importantly, it is against the expectations of the buffering model put forward by Cohen and Wills, as the reduction of stress levels as social support increases is less intense at higher levels of the stressor than at the lower ones. This result is known as the “reverse buffering effect” and has already been described in relation to work stressors [44,45]. The reverse buffering effect occurs either because “excessive” social support undermines individuals’ confidence in their capacity to overcome the stresses or because (given that social support involves communication) sharing negative experiences or perceptions could reinforce stressors [46,47]. The latter hypothesis could be well adapted to the case of employment precariousness, given the tendency to socialize with people from similar socio-economic situations (as theorized in the Introduction) and that closer providers of social support frequently include dependent persons who may represent a further source of concern for the precarious worker. Unfortunately, this finding is based on our small sample size. Therefore, a straightforward recommendation derived from our study for further research is the testing of the buffering hypothesis concerning employment precariousness and social support on a general population representative sample size, something that, to our knowledge, has not been done but is perfectly feasible if a perceived health outcome is chosen. In fact, this approach was used to test the buffering hypothesis on unemployment, although in this case direct and not reverse buffering effects of social support were found [8]. Moreover, the moderate correlation between precariousness and social support suggests that alternative causal relationships between the variables (a mediator role of social support, in particular) could be tested.

Social support proved to be completely unrelated to the set of steroid biomarkers. As for precariousness, it is also unrelated to most of them, including cortisol. These results are in line with [39], which failed to reveal reliable HCC associations with psychosocial variables, and confirm the difficulty of moving from the psychosocial dimension to the biological one in research into the mechanisms through which social conditions affect health. Even biologically grounded variables such as sex, age, and BMI present a weak and unstable association with the biomarkers. It can be concluded there is still much work to be done in order to understand which factors are influencing these cortisol metabolites levels, or even their very meaning in relation to stress, before moving to more sophisticated causal analysis like that attempted here. Other limitations of our research are related to its cross-sectional design, which is prone to unobserved heterogeneity effects between individuals, and the limited sample size, which advises against adjusting for a greater number of variables.

From a positive perspective, and given the former, the significant interactions found between precariousness and sex in predicting levels of the 20αDHE and 20βDHE biomarkers for women must be given value. Analysis in progress suggests that specific dimensions of the EPRES scale are more clearly related to certain biomarkers that are different depending on sex.

## 5. Conclusions

Precarious employment proved to be a risk factor for increased stress. Moreover, perceived social support levels are lower for workers with higher precariousness, which may be due to their weakened social networks. Our results also suggest that social support can intensify rather than buffer the impact of precarious employment on perceived stress, although this effect is small in statistical terms. This could be explained because the sharing of negative experiences or perceptions within the worker’s social support networks could reinforce stressors. These findings could not be replicated when the former analytical framework was applied to a set of steroid biomarkers. Social support was found to be completely unrelated to them, while precarious employment was weakly related to certain biomarkers in the case of women. More research is needed to understand how these social factors may translate into steroid profiles in hair.

## Figures and Tables

**Figure 1 ijerph-19-01909-f001:**
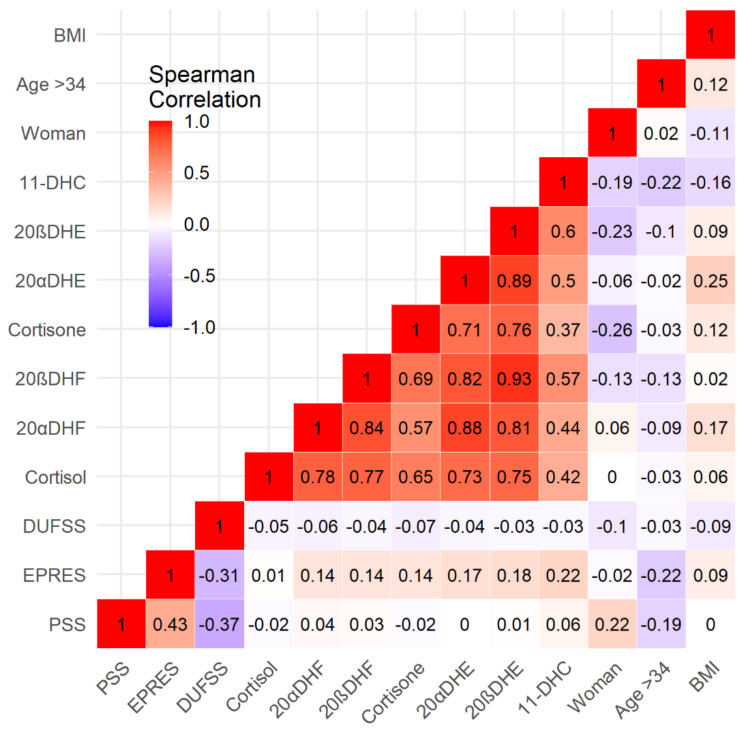
Bivariate Spearman correlations between the variables in study.

**Figure 2 ijerph-19-01909-f002:**
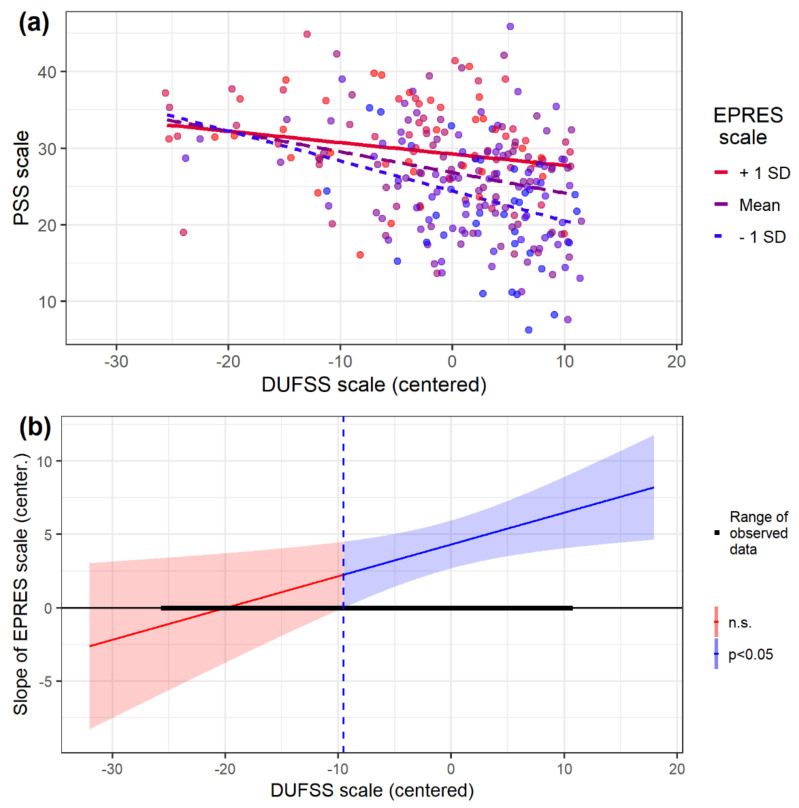
Plot of interaction of EPRES and DUFSS scales (**a**) and region of significance of the EPRES slope (**b**).

**Figure 3 ijerph-19-01909-f003:**
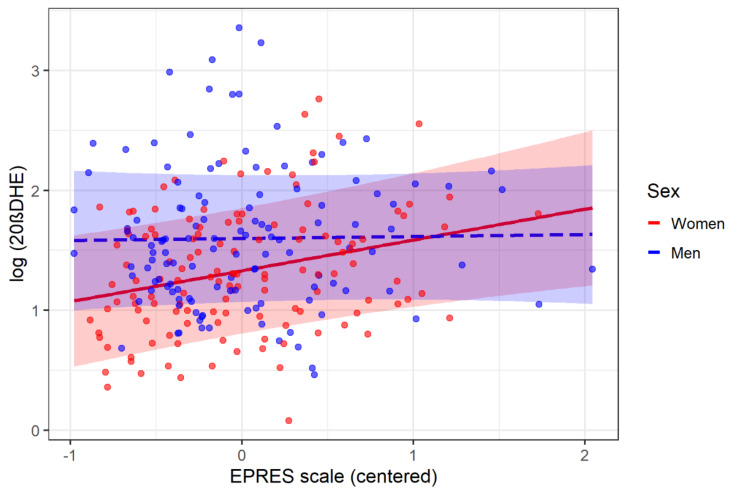
Plot of the interaction effect between sex and EPRES on 20βDHE biomarker.

**Table 1 ijerph-19-01909-t001:** Descriptives of the variables in the research.

Variables	Range	1st Quartil	Median	Mean	3rd Quartil	Missing
Outcomes						
Perceived Stress Scale (PSS scale)	1–44	19.00	24.00	24.34	30.00	0
Cortisol (ng/mg)	1.12–70.27	4.76	6.88	9.75	11.71	4
20α-dihydrocortisol (20αDHF, ng/mg)	0.10–7.60	0.35	0.67	1.01	1.14	5
20ß-dihydrocortisol (20βDHF, ng/mg)	1.01–23.12	2.85	4.05	5.05	5.98	0
Cortisona (ng/mg)	3.08–128.05	19.47	25.89	30.32	35.17	0
20α-dihydrocortisone (20αDHE, ng/mg)	1.57–61.98	5.15	7.17	9.62	11.54	0
20ß dihydrocortisone (20βDHE, ng/mg)	1.31–36.79	3.57	5.02	6.41	7.42	0
A_11dehydrocorticosterona (11-DHC, ng/mg)	0.56–10.08	1.74	2.40	2.77	3.31	1
Predictors						
Precariousness (EPRES scale)	0.06–3.01	0.61	0.96	1.03	1.39	0
Social Support (DUFSS scale)	19–55	41	46	44.48	50.3	0
Adjustment variables						
Woman	0–1	---	---	0.51	---	0
Age > 34 (years)	0–1	---	---	0.74	---	0
Body Mass Index (kg/m^2^)	16.61–42.91	22.23	24.51	25.04	27.17	0

**Table 2 ijerph-19-01909-t002:** Main effects (1) and interaction (2) regression results of social support and employment precariousness predicting perceived stress levels.

	Model (1): Main Effects Only				Model (2): Interaction			
Predictor ^1^	B	CI 95% ^2^		*p*	B	CI 95% ^2^		*p*
		LI	LS			LI	LS	
EPRES	4.17			<0.001	4.32	2.68	5.95	<0.001
DUFSS	−0.23			<0.001	−0.27	−0.38	−0.16	<0.001
DUFSS*EPRES	---				0.22	0.04	0.39	0.014
Woman	3.37			<0.001	3.45	1.80	5.09	<0.001
Age > 34	−2.42			0.012	−2.27	−4.15	−0.38	0.019
Intercept	26.83			<0.001	26.84	23.32	30.35	<0.001
				---				---
Model adjustment								
Observations	255				255			
R^2^	0.289				0.304			
F statistic	F(4;250) = 25.398, *p* < 0.001				F(5;249) = 26.275, *p* < 0.001			

^1^ Mean-centered values of EPRES and DUFSS scales were used. ^2^ A heteroscedasticity consistent standard error (HC3) and covariance matrix estimator were used.

**Table 3 ijerph-19-01909-t003:** Regression results of social support and employment precariousness interaction model predicting different biomarker levels.

	**Outcome ^1^: Cortisol (ng/mg)**					**Outcome ^1^: Cortisone (ng/mg)**			
**Predictor ^2^**	**B**	**CI 95% ^3^**		** *p* **	**Predictor ^2^**	**B**	**CI 95% ^3^**		** *p* **
		LI	LS				LI	LS	
EPRES	−0.05	−0.28	0.17	0.644	EPRES	−0.01	−0.16	0.14	0.909
DUFSS	−0.003	−0.01	0.01	0.544	DUFSS	−0.001	−0.01	0.01	0.779
DUFSS*EPRES	0.01	−0.01	0.03	0.432	DUFSS*EPRES	0.00	−0.01	0.01	0.693
Woman	−0.03	−0.21	0.16	0.754	Woman	−0.23	−0.37	−0.10	0.001
Woman*EPRES	0.05	−0.29	0.40	0.760	Woman*EPRES	0.16	−0.07	0.38	0.172
Age > 34	−0.11	−0.32	0.11	0.326	Age > 34	−0.02	−0.17	0.13	0.781
BMI	0.02	−0.01	0.05	0.195	BMI	0.02	−0.01	0.04	0.172
Intercept	1.75	0.99	2.52	<0.001	Intercept	3.03	2.45	3.60	<0.001
Model adjustment					Model adjustment				
Observations	251				Observations	255			
R^2^	0.017				R^2^	0.076			
F statistic	F(7;243) = 0.547, *p* = 0.7984				F statistic	F(7;243) = 3.272, *p* = 0.002			
	**Outcome ^1^: 20αDHF (ng/mg)**					**Outcome ^1^: 20βDHE ng/mg)**			
**Predictor ^2^**	**B**	**CI 95% ^3^**		** *p* **	**Predictor ^2^**	**B**	**CI 95% ^3^**		** *p* **
		LI	LS				LI	LS	
EPRES	0.01	−0.30	0.32	0.973	EPRES	0.02	−0.15	0.19	0.848
DUFSS	0.002	−0.01	0.02	0.766	DUFSS	0.002	−0.01	0.01	0.675
DUFSS*EPRES	0.01	−0.02	0.03	0.596	DUFSS*EPRES	0.00	−0.01	0.02	0.649
Woman	0.10	−0.12	0.31	0.378	Woman	−0.27	−0.41	−0.13	<0.001
Woman*EPRES	0.29	−0.13	0.70	0.176	Woman*EPRES	0.24	0.01	0.47	0.042
Age > 34	−0.19	−0.46	0.09	0.178	Age > 34	−0.06	−0.23	0.10	0.461
BMI	0.05	0.02	0.08	0.001	BMI	0.01	−0.01	0.03	0.186
Intercept	−1.33	−2.14	−0.52	0.001	Intercept	1.60	1.07	2.13	<0.001
Model adjustment					Model adjustment				
Observations	250				Observations	255			
R^2^	0.081				R^2^	0.105			
F statistic	F(7;242) = 2.742, *p* = 0.009				F statistic	F(7;247) = 5.0667, *p* =< 0.001			
	**Outcome ^1^: 20βDHF (ng/mg)**					**Outcome ^1^: 11-DHC (ng/mg)**			
**Predictor ^2^**	**B**	**CI 95% ^3^**		** *p* **	**Predictor ^2^**	**B**	**CI 95% ^3^**		** *p* **
		LI	LS				LI	LS	
EPRES	0.01	−0.17	0.19	0.906	EPRES	0.11	−0.07	0.28	0.234
DUFSS	0.002	−0.01	0.01	0.738	DUFSS	0.001	−0.01	0.01	0.886
DUFSS*EPRES	0.00	−0.01	0.02	0.683	DUFSS*EPRES	0.003	−0.01	0.02	0.685
Woman	−0.17	−0.31	−0.03	0.015	Woman	−0.17	−0.30	−0.04	0.009
Woman*EPRES	0.18	−0.06	0.41	0.138	Woman*EPRES	0.15	−0.09	0.40	0.214
Age > 34	−0.11	−0.28	0.06	0.219	Age > 34	−0.19	−0.35	−0.03	0.018
BMI	0.01	−0.01	0.03	0.568	BMI	−0.02	−0.04	0.00	0.014
Intercept	1.58	1.04	2.12	<0.001	Intercept	1.84	1.35	2.32	<0.001
Model adjustment					Model adjustment				
Observations	255				Observations	254			
R^2^	0.054				R^2^	0.131			
F statistic	F(7;247) = 2.993, *p* = 0.005				F statistic	F(7;246) = 5.8585, *p* =< 0.001			

^1^ All outcomes have been transformed into logarithms to correct skewness. ^2^ Mean−centered values of EPRESS and DUFSS scales were used. ^3^ A heteroscedasticity consistent standard error (HC3) and covariance matrix estimator were used.

## Data Availability

The data presented in this study are available in anonymized form upon request from the corresponding author. The data are not publicly available due to their containing information that could compromise the privacy of the research participants.

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
