# Peer review of "Precarious Employment and Chronic Stress: Do Social Support Networks Matter?"

_ijerph, 2022, doi:10.3390/ijerph19031909_

Round 1
Reviewer 1 Report
Dear Authors,
I read your paper entitled ,,Precarious Employment and Chronic Stress: Do Social Support Networks Matter?” and I have some suggestions:
In the Abstract, you should mention the time period in which you conducted the research, the novelty, and the usefulness of your research. It would also be more than appropriate to provide additional information on how you collected the data. The objective of the research is also not clear from the Abstract.
The number of keywords is too high. I suggest you to limit them to the most relevant 5.
The Introduction is too long. In addition, I do not recommend such a way to make the Introduction, in subchapters. Try to follow the typical (classic) Introduction to a scientific paper.
Literature Review should also be developed and written in such a way that it reflects exactly what is needed, ie a description of the existing scientific framework to highlight the results already obtained and to capture the gaps in the literature that you want to cover and that underlie research hypotheses. The latter, as well as the objective of the research, should be very clearly highlighted.
The statement in lines 135-136 will disqualify the research I suggest you either give details or give it up. In addition, please explain, with reference to literature, how relevant a sample of 255 people for such research is.
The Methodology does not apply. This must be technically described.
Discussions and Conclusions need to be developed. The Conclusions are irrelevant.
The research is conducted on a sample of 255 people from Spain, more specifically from Barcelona. The sample is not described in detail, and the Discussions and Conclusions (we can ignore them) are general. Such a thing is not possible. Therefore, the argumentative part is weak and the work needs numerous improvements on all its components.
Reviewer 2 Report
I believe this is a well-done study with only minor revisions needed for publication.
In the Introduction, the terms “precarious employment”, first introduced in line 32, and “social support”, first introduced in line 56, should each be clearly defined.
In the Results section, the paragraph from lines 183 to 188 is confusing. Instead of using the term “between them” in lines 184 and 185, a clearer reference should be used. In line 186, a clearer phrase should be used than “the former groups are practically unrelated” (i.e., does this mean that the groups are unrelated to each other?).
Further down in the Results section, lines 210-213 are confusing. What is the difference between “the magnitude of the effect of the precariousness level” and the “beneficial influence of social support”. These terms need to be explained better.
In the Discussion section, the finding that the reduction of stress levels as social support increases is less intense at higher levels of the stressor than at the lower ones needs to be discussed in more depth (since I believe this is the most interesting and important result of the study). As it stands, there is only one sentence (lines 268-270) explaining the result, which is not sufficient. What are the theoretical implications of this finding, e.g., on the main effects model. What would be ideas for future research to explore this relationship further?
Reviewer 3 Report
Manuscript No.: ijerph-1551534
Manuscript Title: Precarious Employment and Chronic Stress: Do Social Support Networks Matter?
Abstract: I suggest briefly presenting the types of methodologies conducted.
Introduction:
-I suggest the authors to write the data in a neutral way, without citing so many previous studies. It reads more like a literature review. In addition, the literature review section is missing.
- I suggest authors write the literature gap clearly in the introduction. Focus must be kept on the background of the problem being explored.
- The authors could write more in the introduction the themes of employees’ perceived employment precariousness especially during the pandemic period. Or what is the key or unique employment precariousness perception during COVID19?
Literature Review:
This section is missing.
I do not see the overview of employment precariousness section, as it is too superficial and lacking in details.
Limitations and Future Research Directions
I believe that cultural aspects or the key characteries on employment relationship during COVID19 need to be brought into this discussion.
In general terms, I recommend authors to review the manuscript, paying attention to the fact that the research results are repeated several times throughout previous materials.
Reviewer 4 Report
I read the article with great pleasure. There are only two major points (and one minor) that need to be amended, before the publication. I will list them below.
- INTRODUCTION: Please provide a brief description of the different stress bio-makres, reported in the study (Cortisol, 149 20α-dihydrocortisol (20αDHF), 20ß-dihydrocortisol (20βDHF), Cortisona, 20α-dihydro-150 cortisone (20αDHE), 20ß dihydrocortisone (20βDHE), and A_11dehydrocorticosterona 151 (11-DHC)) The questionaires are cleary descibed, but the biomarkes no so much.
- SAMPLE - in the line 135-136 You write: "Further details of the sampling design and other aspects of the protocol research may be found elsewhere" - I believe a more robust sample description is needed within the current article, plus a disticition of what is new in current article, and what is already reported in the referenced RESEARCH PROTOCOL is neccesary (else the article is reporting already published research).
- Method: Please describe why the Johnson-Neyman method was used? What is the added benefit of this analysis?
Else, the study is great and can be published.
Round 2
Reviewer 1 Report
Dear Authors,
Your effort to improve the work is obvious.
I suggest you to make the effort to respect the previous requirements that you did not check, especially insisting on the technical methodological aspects and the development of the conclusions.
Success!
Reviewer 3 Report
Manuscript No.: ijerph-1551534
Manuscript Title: Precarious Employment and Chronic Stress: Do Social Support Networks Matter?
The manuscript has been substantially reformatted and modified based on reviewers’ suggestions and the conclusions have been amended.
I suggest authors could prepare a revision note that comprises of the changes incorporated in response to the comments provided by reviewers. It should clearly outline the reviewer’s responses and includes page and line numbers where the relevant revision is incorporated.
The literature review part is still missing.
I do not understand the meaning of “This process particularly affects the most vulnerable social groups in the labour market, like women, immigrants, the working class and young workers”.
In my opinion the authors miss the most likely explanation for gap in social epidemiological research.
